# The Acute Phase Reaction and Its Prognostic Impact in Patients with Head and Neck Squamous Cell Carcinoma: Single Biomarkers Including C-Reactive Protein Versus Biomarker Profiles

**DOI:** 10.3390/biomedicines8100418

**Published:** 2020-10-14

**Authors:** Helene Hersvik Aarstad, Svein Erik Emblem Moe, Øystein Bruserud, Stein Lybak, Hans Jørgen Aarstad, Tor Henrik Anderson Tvedt

**Affiliations:** 1Department of Clinical Science, Faculty of Medicine, University of Bergen, 5021 Bergen, Norway; Helene.Aarstad@uib.no (H.H.A.); oystein.bruserud@helse-bergen.no (Ø.B.); 2Department of Otolaryngology/Head and Neck Surgery, Haukeland University Hospital, 5021 Bergen, Norway; svein.erik.emblem.moe@helse-bergen.no (S.E.E.M.); stein.lybak@helse-bergen.no (S.L.); 3Section for Hematology, Department of Medicine, Haukeland University Hospital, 5021 Bergen, Norway; tor.henrik.anderson.tvedt@helse-bergen.no; 4Department of Clinical Medicine, Faculty of Medicine, University of Bergen, 5021 Bergen, Norway

**Keywords:** head and neck squamous cell carcinoma, C-reactive protein, interleukin 6, ciliary neurotrophic factor, interleukin 1, interleukin 31, interleukin 33, tumor necrosis factor α

## Abstract

C-reactive protein (CRP) has a prognostic impact in head and neck squamous cell carcinoma (HNSCC). However, the acute phase reaction involves many other proteins depending on its inducing events, including various cytokines that can function as reaction inducers. In the present study, we compared the pretreatment acute phase cytokine profile for 144 patients with potentially curative HNSCC. We investigated the systemic levels of interleukin (IL)6 family mediators (glycoprotein (gp130), IL6 receptor (R)α, IL6, IL27, IL31, oncostatin M (OSM), ciliary neurotrophic factor (CNTF)), IL1 subfamily members (IL1R antagonist (A), IL33Rα), and tumor necrosis factor (TNF)α. Patient subsets identified from this 10-mediator profile did not differ with regard to disease stage, human papilloma virus (HPV) status, CRP levels, or death cause. Increased CRP, IL6, and IL1RA levels were independent markers for HNSCC-related death in the whole patient population. Furthermore, gp130, IL6Rα, and IL31 were suggested to predict prognosis among tumor HPV-negative patients. Only IL6 predicted survival in HPV-positive patients. Finally, we did a clustering analysis of HPV-negative patients based on six acute phase mediators that showed significant or borderline association with prognosis in Kaplan–Meier analyses; three subsets could then be identified, and they differed in survival (*p* < 0.001). To conclude, (i) HPV-negative and HPV-positive HNSCC patients show similar variations of their systemic acute phase profiles; (ii) the prognostic impact of single mediators differs between these two patient subsets; and (iii) for HPV-negative patients, acute phase profiling identifies three patient subsets that differ significantly in survival.

## 1. Introduction

The acute phase reaction is characterized by increased levels (defined as >25% increase) of various serum proteins in response to inflammation or tissue injury [1,2], but the name is misleading because it can accompany both acute and chronic states, including cancers [3]. The reaction is induced by or includes various cytokines released during inflammation, including interleukin (IL) 6 family cytokines, IL1 subfamily mediators (IL1α/β, IL1 receptor antagonist (IL1RA), soluble IL33Rα chain), and tumor necrosis factor α (TNFα) [1,4,5,6,7]. Inflammatory markers (e.g., acute phase proteins and especially C-reactive protein (CRP), leukocytosis, thrombocytosis) are associated with prognosis in several malignancies, including head and neck cancer/squamous cell carcinoma (HNSCC) [8,9,10]. Thus, cytokines can be classified as acute phase proteins and thereby contribute to the prognostic impact of inflammatory biomarkers in human cancers.

Immunotherapy has recently been approved for HNSCC [11], and host inflammatory responses may be involved in HNSCC carcinogenesis [12]. Soluble inflammation-regulatory mediators (e.g., cytokines, soluble cytokine receptors) can then be important for the communication between immunocompetent cell subsets or between immunocompetent cells and cancer cells [13]. The inflammatory mediators may therefore have local effects in the microenvironment of the releasing cells or hormone-like effects on distant organs [14,15]; in both cases, the effects of single cytokines can be influenced by other mediators through the local or systemic cytokine network [16].

IL6 is produced by many different cells, e.g., macrophages, T and B cells, endothelial cells, adipocytes, and various cancer cells [13]. It can promote differentiation of immunocompetent cells, induce the acute phase reaction, stimulate hematopoiesis [17,18,19], and be involved in carcinogenesis as well as the development of cancer-associated cachexia [13]. The systemic (i.e., serum or plasma) IL6 levels at diagnosis seem to have a prognostic impact in many human malignancies [20], including head and neck cancer [21,22,23,24]. IL6 mediates its effects through two main mechanisms. In classical signaling, IL6 binds to its membrane-bound IL6Rα/glycoprotein(gp)130 receptor that is expressed by few cell types (hepatocytes, neutrophils, monocytes, macrophages, some lymphocytes) [14], whereas in trans signaling, the soluble IL6/IL6Rα complex binds to membrane-bound gp130 [25]. gp130 is the signal-transducing IL6 receptor chain, which is expressed by a wide range of cells [25,26] and can activate downstream Janus kinase/Signal transducers and activators of transcription (JAK-STAT), Ras-Extracellular signal-regulated kinases (ERK), and Phosphatidylinositol 3-kinase/Protein kinase B (PI3K-Akt) signaling pathways [12]. On the other hand, soluble gp130 serves as a decoy receptor of the IL6 reservoir [27]. Finally, IL6 belongs to the IL6 cytokine family [14,28], which also includes IL11, IL27, IL31, ciliary neurotrophic factor (CNTF), leukemia inhibitory factor (LIF), oncostatin M (OSM), and cardiotrophin-like cytokine (CLC) factor 1 [29]. These cytokines share gp130 as the signal-transducing part of their receptors and several of them can mediate their activity both through classical and gp130-mediated trans signaling [29].

The IL1 cytokine subfamily includes IL1 and IL33 [30]. The IL33Rα chain exists in membrane-bound and truncated soluble (i.e., a decoy receptor) isoforms [31]. Membrane-bound IL33Rα is expressed both by epithelial, immunocompetent, and hematopoietic cells [31,32], and the truncated isoform can thereby be released by various cells [31]. In contrast, IL33 is expressed mainly by non-hematopoietic cells [32], and it can be constitutively expressed and inactivated by several proteases. It is classified as an alarmin released in response to cell damage [32]. IL33 binding to IL33Rα recruits a co-receptor (usually IL1RAcP, the same as the IL1 receptor [31]) that initiates the activation of various downstream pathways [31,32]. IL33 can also regulate transcription through direct binding to heterochromatin or binding and thereby inactivation of nuclear factor kappa B (NF-κB) [32,33,34,35]. Recent studies suggest that IL33/IL33Rα is involved in HNSCC carcinogenesis, possibly through both direct effects on the cancer cells and indirect effects via stromal/immunocompetent cells [36,37,38,39]. Finally, IL1RA is another member of the IL1 subfamily that is secreted both by immunocompetent and epithelial cells [30]. It blocks the receptor of pro-inflammatory IL1α/IL1β [30], two mediators that can be involved in human carcinogenesis through their effects on stromal cells [40].

One important matter regarding HNSCC is the dual possible causes of primarily oropharyngeal SCC, i.e., the classical causes of smoking and alcohol versus the human papilloma virus (HPV) [41,42]. The latter has become evident as a risk factor for the disease during the last decades [43]. These two entities were not clinically separated before PCR techniques developed but have eventually been shown to be of clinical importance [44]. The study of this distinction has been a very important task, especially focusing on de-toxification of treatment [45] and precision medicine [46].

Systemic inflammatory responses, including the systemic levels of acute phase cytokines, have a prognostic impact in several cancers, including HNSCC [9,10,21,22,23,47,48,49,50,51,52,53,54,55,56,57]. However, the acute phase reaction is an example of how several cytokines (including IL6 family and IL1 subfamily members as well as TNFα) form an interacting network [1]. In this context, we investigated this acute phase cytokine profile for a large group of HNSCC patients at the time of diagnosis. These mediator levels were compared with healthy controls and subsequently examined by unsupervised hierarchical clustering analysis in order to reveal any differences in the cytokine patterns among subgroups of cancer patients. In particular, the HNSCC group was divided according to tumor HPV status.

## 2. Materials and Methods

### 2.1. Participants in the Study

The study was approved by the Regional Ethics Committee (Committee for Medical and Health Research Ethics, referred to as REK Vest) with project identification codes REK 2011-125 (approved date 1 March 2011), REK 2011/520 (approved date 4 April 2011), and REK 2014/2348 (approved date 30 January 2014). The study was performed in accordance with the Declaration of Helsinki. All samples were collected after written informed consent. The study included 144 consecutive patients newly diagnosed with head and neck squamous cell carcinoma and admitted to Department of Otolaryngology/Head and Neck Surgery, Haukeland University Hospital (Bergen, Norway) during the years 2013 until 2018. The patients were scheduled for curative treatment, and those with autoimmune disease or on systemic corticosteroid therapy were not included. The characteristics of the patients are given in Table 1. In addition, 15 presumably healthy individuals were also included in the study.

### 2.2. Laboratory Analyses

The biopsy analyses for HPV tumor infection have been described in detail previously [59,60]. CRP serum levels was analyzed using an immunoturbidimetric method provided by Roche (Basel, Switzerland), and during the entire period, the lower limit of detection for CRP was 1 mg/L. Standard laboratory values like the hematological analyses of hemoglobin, leukocyte, and thrombocyte counts were also performed.

At arrival during the first HNSCC hospital admission, a peripheral venous blood sample was drawn before any cancer-specific treatment had started. Samples were prepared by gradient centrifugation with Lymphoprep^®^ and plasma was collected, aliquoted, and later stored frozen at −80 °C until analysis. Later, the samples were thawed and centrifuged at 16,000× *g* for 4 min immediately prior to a further procedure. IL33Rα and IL1RA levels were analyzed by using Luminex analyses (R&D Systems Europe Ltd., Abingdon, UK): Human Cardiac Base Kit A and Human Premixed Multi-Analyte Kit, respectively. The latter kit from R&D Systems was also used to quantify most of the IL6 family mediators (gp130, IL6Rα, IL27, IL31, and OSM as well as TNFα), whereas CNTF was analyzed by the Human Pituitary Magnetic Bead Panel 1 (EMD Millipore Corporation, Billerica, MA, USA). IL6 levels were determined by a high-sensitivity IL6 and IL1β by Human High Sensitivity Cytokine Base Kit B (R&D Systems). All analyses were performed strictly according to the manufacturer’s instructions and the levels estimated by using the Luminex^®^ 100^TM^ System (Luminex Corporation, Austin, TX, USA). In all these antibody-based assays, estimation of plasma levels was based on comparison with standard curves (six or seven standards included in all assays and for all cytokines) that were prepared according to the manufacturers’ instructions and with standard samples supplied by the manufacturer. All kits are suitable for analysis of plasma samples (manufacturers’ information). All standards and samples were analyzed in duplicates and the mean level of duplicates used in all analyses. IL6 values were available for all 144 patients, but the other mediators were measured in 143 patients due to technical reasons. For the same reason, one additional patient was lacking for CNTF analysis.

To evaluate inter-plate variation, we included one patient sample in all assays without detecting substantial differences between them. The variation between duplicates was generally <10% of the mean concentration. In neither case did the cytokine/receptor correlation coefficients exceed 0.30 for any mediator versus sample storage time.

### 2.3. Statistical and Bioinformatical Analyses

The IBM^®^ SPSS^®^ Statistics software, version 25.0 (IBM Corp., Armonk, NY, USA) was utilized. Comparison of categorized descriptive data was performed by cross-tables and the exact Chi-square test. Kendall’s tau (τ) was used for correlation analyses and Mann–Whitney U-tests for comparison of different groups. The rationale for these non-parametric choices was the predominantly not normally distributed data. Mediator levels were presented as the median and the variation range. Regarding survival, the cytokines/receptors were either dichotomized by the median (CRP, IL6, IL1RA, IL6Rα) or quartiles (gp130, IL31). Kaplan–Meier analyses were used for the percentage estimation of outcome prediction, including a Log-Rank test between groups. Cox proportional hazard models were also used for survival analyses, and predictions from various mediators generally given as a hazard ratio, 95% confidence interval, and *p*-value. A *p*-value < 0.05 was regarded as statistically significant. Multiple comparisons were taken into account and corrected with Bonferroni, in that the chosen significance level was divided by the number of comparisons in the respective analyses. Five-year patient survival was determined from the Norwegian Population Registry by the end of March 2020. In total, 117 out of the 144 patients were then alive.

Cut-off mediator values regarding sensitivity and specificity were examined among the HPV-negative subgroup by a Receiver operating characteristic (ROC) curve with coordinate points.

Bioinformatical analyses were performed by the use of J-Express (MolMine AS, Bergen, Norway) [61]. All cytokine and receptor levels were normalized by their median value, log(2)-transformed, entered into a complete linkage, and a hierarchical clustering generated. Distance measures were Euclidean.

IL6 levels were determined for all 144 patients, CNTF levels were not available for two patients, and for the other cytokine mediators, levels were available for 143 patients. The median of the patient cohorts was used for these exceptional patients in the bioinformatical analyses.

## 3. Results

### 3.1. Patients Included in the Study

We included 144 consecutive patients (105 men, 39 women) in our study, all intended for curative treatment (Table 1, Appendix A). The median age was 62 years, and the majority of them were males. The patients showed wide variations in peripheral blood leukocyte (i.e., white blood cell, WBC) counts and thrombocyte counts as well as serum C-reactive protein (CRP) levels. The HPV status was investigated for 102 patients; 57 patients (40%) were then HPV positive and 45 patients were HPV negative. The remaining 42 patients were not tested but probably had HPV-negative cancers because they had primary tumors localized outside the oropharynx site [58]. For these reasons, the patients were classified in two main subgroups, i.e., HPV-negative plus unknown HPV-status versus HPV positive, respectively (see Table 1). Most of the 57 HPV-positive patients had tumors positive for HPV16 (48 patients), but other subtypes detected were HPV18 (3 patients), HPV33 (3 patients), HPV35, HPV58, and HPV73 (one patient each). The clinical and biological characteristics of patients with HPV-positive tumors differed from the other patients; as expected, the majority of the HPV-positive cancers were localized to the tonsils or tongue base (50 out of 57 patients), a majority of them had stage T1/T2 tumors (42/57), and the N stage for a majority of patients was N2 (31/57).

### 3.2. The Acute Phase Profile of the Cancer Patients Differs from Healthy Controls

We compared the systemic cytokine levels for the 144 cancer patients and a group of 15 healthy individuals (Appendix A). All cytokine/cytokine receptor/cytokine antagonist levels were measured by antibody-based technologies (see Section 2.2). The levels of IL6 (all 144 patients tested), IL27 (143 patients analyzed), IL31 (143 patients), CNTF (142 patients), IL1RA (143 patients), and TNFα (143 patients) were significantly altered in the patients; all of them were increased except for CNTF, which was significantly decreased in the patients compared with the healthy controls. The clearest difference was observed for IL6; for this cytokine, the *p*-value remained significant even after Bonferroni correction, and most of the patients showed plasma levels above the variation range of the healthy controls (Appendix A). The other measured mediators did not differ significantly between the two groups. Finally, IL1β levels were undetectable for most patients and were not included in the following statistical analyses.

### 3.3. The Acute Phase Reaction Differs between Patients with Head and Neck Squamous Cell Carcinoma

The 144 patients showed a wide variation in CRP levels at the time of diagnosis. The CRP level showed a highly significant correlation with tumor stage (*p* = 0.001) and an association of borderline significance with smoking (*p* = 0.021), whereas no statistically significant associations were observed between CRP levels and age or sex (Appendix A). Thus, patients with head and neck squamous cell carcinoma are heterogeneous with regard to the development of an acute phase reaction (i.e., show wide variation in CRP levels), and the induction of an acute phase reaction depends both on clinical/life style and tumor characteristics.

We investigated the possible correlations between CRP, peripheral blood cell counts, and the plasma levels for the 10 soluble acute phase mediators (Appendix A). Although several associations were detected based on the *p*-values (Kendall’s test, *p*-value < 0.01), the Kendall’s τ was below 0.30 for most of the combinations (Appendix A). Relatively strong associations were detected for CRP-WBC count (τ 0.30), CRP-IL6 (0.47) and WBC-IL6 (0.34); additional relatively strong correlations were only seen for gp130-IL31 (0.30), IL6Rα-OSM (0.37), and IL27-TNFα (0.30). Taken together, these results suggest that the CRP-WBC-IL6 triad reflects similar characteristics of the acute phase reaction, whereas the lack of strong correlations between the various soluble mediators reflects a heterogeneity between the cancer patients with regard to the acute phase cytokine response.

### 3.4. Two Main Patient Subsets Were Identified Based on the Plasma Profile of IL6 Family Mediators

We did a hierarchical clustering analysis only based on the plasma levels of seven IL6 family mediators in our 144 patients (deviations from this number in parenthesis): IL6 (levels available for all patients), gp130 (measured in 143 patients, see Section 2.2), IL6Rα (143), OSM (143), IL31 (143), IL27 (143), and CNTF (142 patients analyzed). This analysis showed that the patients could be divided into two main subsets, i.e., an upper/left subset including 99 patients and a lower/right subset including 45 patients (Figure 1). The clinical and biological characteristics of the individual patients according to the cluster group can be found in Appendix A. As can be seen from the comparisons presented in Table 2, these two patient subsets showed highly significant differences (*p*-values < 0.05/14 comparisons, i.e., also significant after Bonferroni correction) for IL6, IL31, and CNTF, whereas the differences in the IL6Rα (*p* = 0.039), IL1RA (*p* = 0.027), and TNFα (*p* = 0.016) levels between the two clusters reached only borderline significance (0.05 > *p*-value > 0.05/14). Thus, the overall impact from other IL6 family members and not only IL6 alone seems to be more decisive for the identification of these two main patient subsets based on this IL6 family clustering.

The clustering analysis presented in Figure 1 was only based on the systemic levels of IL6 family mediators, and the sub-classification of the patients is thus dependent on the IL6 family mediators alone. However, IL1RA and TNFα also differed between these two identified patient subsets although these mediator differences reached only borderline significance, whereas the IL33Rα levels did not show any statistically significant difference between the two patient subsets (Table 2).

Even though we did not include CRP, hemoglobin levels, WBC, and platelet counts in the IL6 family clustering analysis (Figure 1), we compared these levels for the two patient subsets identified by this hierarchical clustering analysis (Table 2). The CRP levels showed a highly significant difference between these two patient subsets, whereas neither the hemoglobin levels, WBC counts, nor platelet counts differed significantly between the two main patient subsets (Table 2). These observations are consistent with the results for the correlation analyses (Appendix A), where only CRP levels showed a correlation with IL6 levels.

We also compared the mediator levels for each of the two main clusters with the levels in healthy controls (Appendix A). For the 99 patients in the upper main cluster, only IL6 showed a highly significant difference (*p* < 0.001) that remained significant after Bonferroni correction, while IL31 (*p* = 0.025) and ILRA were less significant (*p* = 0.023). In contrast, the 45 patients in the lower main cluster showed highly significant differences compared with healthy controls both for IL6, IL31, CNTF, IL1RA, as well as TNFα; they showed less significant differences for IL27 (*p* = 0.015) and IL33Rα (*p* = 0.026), and a difference of borderline significance for IL6Rα (*p* = 0.040). The presence of the differences in IL6, IL31, CNTF, IL1RA, and TNFα remained significant following Bonferroni correction. Thus, especially patients in the lower main cluster show more extensive differences in the plasma IL6 family profile, both when compared with the other patients and with the healthy controls than the patients in the upper cluster.

Despite their differences in the overall acute phase cytokine profile, the two main patient clusters did not differ when comparing the frequencies of patients with HPV-positive tumors, patients above 70 years of age, tumor stage, lymph node metastases, and advanced disease (i.e., advanced tumors and/or lymph node metastases) (Appendix A). Finally, 27 patients died during follow-up (22 from cancer progression), corresponding to 17 out of the 99 patients in the upper cluster and 10 out of the 45 patients in the lower cluster. The two subsets did not differ with regard to the frequency of survivors (Appendix A).

### 3.5. Two Main Patient Subsets are Also Identified Based on an Extended Acute Phase Cytokine Profile

We also did a hierarchical clustering analysis based on the plasma levels of seven IL6 family members (IL6, IL6Rα, gp130, IL27, IL31, OSM, CNTF) together with the two IL1 subfamily mediators (IL1RA, IL33Rα) and TNFα; IL1β was not included in this analysis because these levels were not detectable for most patients (Figure 2). Thus, in contrast to the analysis presented in Figure 1, this second unsupervised hierarchical clustering analysis was based on the impact of 10 different mediators, i.e., including three different and generally accepted inducers of the acute phase reaction (IL1/IL6/TNFα) and not only IL6 and its family members. This analysis also showed that the patients could be divided into two main subsets that were different from the two main clusters observed in the analysis based on the IL6 family mediators alone (see Section 3.4 and Figure 1). The cluster in this 10-mediator analysis identified an upper main patient cluster including 66 patients (left column), a lower main cluster including 77 patients (right column), and one outlier located at the lower end of the clustering diagram (left column). The two main clusters showed highly significant differences that remained significant even after Bonferroni correction (i.e., *p*-values < 0.05/14 comparisons) for IL6Rα, IL31, CNTF, and TNFα. In contrast, differences of borderline significance (0.05 > *p*-value > 0.05/14) were seen for gp130 (*p* = 0.023) and IL27 (*p* = 0.007) (Table 3). In contrast, IL6 levels as well as the CRP levels, WBC counts, platelet counts, or hemoglobin levels did not differ between the two main subsets (Table 3); this is expected from the correlation analyses presented in Appendix A.

We also compared the mediator levels for each of the two main clusters with the levels in healthy controls (Appendix A). The patients in the upper/left main cluster showed a highly significant difference compared with healthy controls only for IL6, which remained significant with Bonferroni correction. A less significant difference was shown for IL1RA (*p* = 0.008). In contrast, for the 77 patients in the lower/right main cluster, both IL6, IL27, IL31, as well as CNTF and TNFα showed highly significant differences (as defined by Bonferroni) compared with the controls, whereas they showed less significant differences for IL6Rα (*p* = 0.030), IL33Rα (*p* = 0.046), and IL1RA (*p* = 0.012). Thus, patients in the lower main cluster show extensive differences in acute phase cytokine profiles, whereas patients in the upper cluster share many similarities with healthy individuals.

Despite these differences in the overall acute phase cytokine profile, the two main patient clusters did not differ when comparing the frequencies of patients with HPV-positive tumors, patients above 70 years of age, T stage, or the frequency of patients with advanced disease as defined in Appendix A. The survival did not differ between the two clusters either; 27 patients were dead during follow-up (22 of them due to cancer progression), corresponding to 15 out of the 67 patients in the upper cluster and 12 out of the 77 patients in the lower cluster (Appendix A). Appendix A shows the clinical and biological characteristics for each patient in the two clusters.

### 3.6. HPV-Positive Patients Show a Similar Variation in Their Systemic Mediator Profile but Differ in Survival Compared with HPV-Negative Patients

A relatively large subset of patients with HNSCC have HPV-positive tumors. As described in a recent review [58], the tumors of these patients are mainly localized in the tonsils or similar oropharyngeal sites, and HPV-positive patients have a better prognosis. Thus, both the predominant tumor localization and the prognosis differs from other HNSCC patients. We therefore did a comparison of patient survival for our HPV-positive patients, HPV-negative patients, and the patients with unknown HPV status but expected to comprise mainly negative patients based on tumor localization [58]. The survival of these three groups differed significantly; the HPV-positive patients had a high survival, whereas HPV-negative and HPV-unknown patients showed a similar decreased survival (Appendix A). This was the case for both HNSCC (disease)-specific (*p* = 0.026) and total survival (*p* = 0.018). However, despite these differences in survival, HPV-positive patients did not differ in their acute phase cytokine profile neither with regard to the levels of individual cytokines, except IL6Rα (Appendix A), or with regard to the overall profiles (Figure 1 and Figure 2).

### 3.7. Single Acute Phase Cytokines Are Associated with Prognosis in Head and Neck Squamous Cell Carcinoma Even Though the Overall Cytokine Profile Shows No Such Associations

We analyzed the prognostic impact (cancer-specific survival, i.e., from HNSCC) for CRP and for each of the 10 cytokine mediators. Twenty-seven patients died during follow-up; 22 patients died from their malignant disease, whereas 5 patients died from other causes. We first investigated the whole patient cohort with regard to HNSCC-specific survival; CRP (hazard ratio (HR): 10.9, 95% confidence interval (CI): 2.5–46.5, Cox *p*-value 0.001) together with IL6 (HR: 11.7, 95% CI: 2.7–50.1, Cox *p*-value 0.001) showed significant predictions (Figure 3). The IL1 subfamily member IL1RA also demonstrated prediction of disease survival (HR: 2.8, 95% CI: 1.1–7.1, Cox *p*-value 0.035). The other IL6 family cytokines, IL1 subfamily mediators, and TNFα showed no significant associations with cancer-specific survival (data not shown). Furthermore, the significance of IL6 was maintained in a multivariate analysis that included IL6 (HR: 7.1, 95% CI: 1.6–32.0, *p*-value 0.011), age (*p*-value 0.723), and T stage (HR: 1.9, 95% CI 1.3–2.9, *p*-value 0.001). The results were similar with additional inclusion of HPV (IL6: *p*-value 0.021, HR 6.1, 95% CI 1.3–28.5). Thus, IL6 as a single mediator, but neither the IL6 cytokine profile nor our extended acute phase cytokine profile, has a prognostic impact in patients with HNSCC. Prognostic prediction for CRP was also demonstrated in a multivariate analysis with the same variables included (HR: 6.2, 95% CI: 1.3–28.7, *p*-value 0.020). When combined into the same model in addition comprising the co-variates age and HPV, both IL6 (HR: 4.7, 95% CI 1.0–22.5, *p*-value 0.049) and CRP (HR: 5.6, 95% CI 1.2–26.0, *p*-value 0.028) predicted HNSCC-specific survival.

We then analyzed the cancer-specific survival for HPV-negative patients, i.e., all patients tested to be negative plus patients not tested but with the primary site of their cancer outside the typical locations for HPV-positive tumors (Figure 4). Again, CRP (HR: 9.7, 95% CI: 2.2–42.1, Cox *p*-value 0.002) and IL6 (HR: 8.4, 95% CI: 1.9–36.4, Cox *p*-value 0.004) showed significant associations with prognosis in univariate analyses; the same was true for IL1RA (HR: 3.3, 95% CI, *p*-value 0.024). In addition, gp130 predicted HNSCC-specific survival (*p* = 0.045), whereas IL6Rα (*p* = 0.066) and IL31 (*p* = 0.052) were indicated by borderline trend results (Figure 5). The other cytokine/receptor mediators did not show any significant associations with cancer-specific survival (data not shown). Furthermore, the significance was maintained in multivariate analyses for CRP (HR: 6.0, 95% CI: 1.2–29.4, *p*-value 0.026), as well as for IL6 (HR: 5.2, 95% CI: 1.1–25.8, *p*-value 0.042). Thus, again, we observed that single acute phase biomarkers are associated with prognosis rather than the cytokine profiles, and the high CRP/IL6 phenotype is associated with prognosis for this patient subset.

We finally analyzed HNSCC-specific survival for HPV-positive patients, i.e., all patients tested to be positive. In Kaplan–Meier analyses, only IL6 (*p* = 0.036) showed a significant association with cancer-specific survival (Figure 4), whereas CRP (*p* = 0.075) and CNTF (*p* = 0.084) demonstrated borderline values. None of the other cytokine mediators reached statistical significance. The low number of cancer-related deaths in this patient subset limited the possibilities of more extensive statistical analyses, but it is justified to suggest that the prognostic impact of acute phase cytokines differs between HPV-positive and HPV-negative patients with regard to single mediators.

### 3.8. HPV-Negative Patients Differ in Survival Based on a Modified Acute Phase Profile from Indicated Predictions

We performed a final clustering analysis for the HPV-negative patients (n = 87) based on the six mediators (i.e., CRP, IL6, IL6RA, gp130, IL6Rα, IL31) showing significant or borderline associations with survival in the Kaplan–Meier analyses above (Figure 6). Three different patient subsets could be identified in this analysis, and in the largest cluster including most patients (n = 46), these generally showed low levels of all six mediators. The three identified patient clusters differed significantly with regard to survival (Figure 7; Kaplan–Meier analysis, *p* < 0.001); the highest survival was seen for the majority of patients in cluster 2 with generally low systemic levels of all six acute phase mediators. A ROC curve was further generated based on the significant predictions from CRP, IL6, and IL1RA to demonstrate the prognostic value of these parameters (Figure 8). Regarding IL6, such analyses determined that an IL6 value below 2.9 pg/mL showed an 80% likelihood of survival versus values above 2.0 pg/mL demonstrating an 80% chance of cancer-related death. The corresponding cut-off values for CRP and IL1RA were 5.5/3.5 mg/L and 644/357 pg/mL, respectively.

## 4. Discussion

The acute phase reaction refers to an increase in the concentrations of serum proteins in response to inflammation or tissue injury, and these proteins are referred to as acute phase proteins [2]. This reaction can accompany both acute and chronic states, including cancers [1]. In the present study, we investigated acute phase cytokines in patients with recently diagnosed HNSCC. Even though the patients showed altered levels of acute phase cytokines, they could be divided into two main subsets based on their acute phase cytokine profile including 10 biomarkers. One subset showed only minor differences in their profiles compared with the healthy individuals, whereas the other subset showed more extensive differences involving both IL6 family cytokines, IL1 subfamily mediators, and TNFα. In survival analyses, cancer-specific prognostic impact was found for CRP, IL6, and IL1RA.

Several previous studies have shown that both increased systemic CRP levels [8,47,48,49,50,51,52] and IL6 levels [9,21,22,23,53,54,55,56,57] are associated with an adverse prognosis in HNSCC. IL6 can function as a driver of the acute phase reaction, i.e., increased CRP levels can at least partly be due to an IL6 effect [1,26]. However, IL6 is only one out of several members of the IL6 cytokine family, a group of cytokines with several common biological characteristics [1,26]. They can initiate intracellular signaling both through cytokine binding to the complete membrane-bound receptor (i.e., the cytokine-binding receptor chain)/gp130 complex or through binding of the soluble cytokine/receptor complex to membrane-bound gp130, certain receptors can bind more than one IL6 family cytokine, and different receptors initiate activation of the same downstream pathways [28,63]. For this reason, we investigated the systemic levels of various IL6 family members as part of the acute phase/cytokine response. However, several other cytokines are also regarded as parts of and/or possible drivers of the acute phase response [1], and for this reason, we also included the IL1 family mediators IL1β (low or undetectable levels in most patients), IL1RA, and IL33Rα together with TNFα. IL33 was not included because it is expressed by the cancer cells, and we therefore regard local release as more important than distant release [36,37].

There are extensive interactions and associations between various cytokines and their cellular responses, including the intracellular crosstalk between signaling pathways downstream to various cytokine receptors, as well as the coordinated release of different cytokines in cytokine responses. The cytokine system is therefore often referred to as the cytokine network. This is also true for the IL6 family/IL1 subfamily/TNFα cytokines investigated in the present study. First, the main downstream pathway for the IL6 family/gp130 is JAK-STAT3, but the Mitogen-activated protein kinases (MAPK)/ERK and the PI3K-Akt-mammalian target of rapamycin (mTOR) pathways are also activated [64,65]. Second, IL1/IL33 both recruit the IL1Rα1 co-receptor and the NF-κB complex is a major downstream target for IL1/IL33 receptor ligation together with JNK, ERK, and p38 [66,67]. Third, the NF-κB complex is also an important downstream intracellular target for TNFα together with Akt and p38 [68,69]. These cytokine receptors thus have a partial overlap with regard to the downstream targets of their receptors. Finally, there may also be a coordinated release for several of these cytokines, e.g., the release of IL1/IL6/TNFα by monocytes activated in response to Toll-like receptor ligation, because such monocyte activation will induce a broad cytokine release response [70]. It is not known whether such a chemokine response will influence or modify the overall biological effects of acute phase reactions.

Stromal carcinoma-related fibroblasts are the main type of non-immune cells in the microenvironment of HNSCC, and these fibroblasts generate mediators through which they interact with cancer cells [36,38]. Studies of global gene expression profiles have identified IL33 as a critical mediator in fibroblast-induced invasiveness in these malignancies. This cytokine induces epithelial-to-mesenchymal trans-differentiation with increased IL33 gene expression in the cancer cells; this is possibly the mechanism behind the correlation between IL33 expression in malignant and stroma cells and between high IL33 expression and high TNM stage/poor prognosis. This conclusion is further supported by another immunohistochemical study that included 81 patients with SCC of the tongue; patients with high IL33 or IL33Rα expression then had significantly worse prognosis, and high IL33 expression was also associated with increased micro-vessel density, i.e., increased density of endothelial cells, which may also be a source of IL33 [37]. Furthermore, IL33 also promotes Treg proliferation in non-lymphoid organs, and a recent immunohistochemistry study therefore investigated IL33^+^ and FoxP3^+^ cells in 68 laryngeal SCC patients [39]. The level of stromal IL33 was significantly upregulated in advanced versus early stage patients, and positively correlated with Foxp3^+^ Treg infiltration as well as a poor prognosis. The infiltrating IL33Rα-expressing Tregs were IL33 responsive, and IL33 improved their suppressive functions. Thus, locally released IL33 may support the development of HNSCC both directly through stimulation of cancer cell proliferation and indirectly though pro-angiogenic effects as well as suppression of potentially anticancer immune reactivity. IL33 should also be regarded as an acute phase cytokine, and for these reasons, we included IL33Rα in our present studies of acute phase cytokine profiles. However, the systemic IL33Rα levels did not show an association with prognosis, unlike what was the case in a recent study on renal cell carcinoma [71].

Inflammatory parameters are considered as possible prognostic markers in several human malignancies [9,55]. These parameters include both acute phase proteins and peripheral blood concentrations of various leukocyte subsets and platelets [62,72,73,74,75]. Increased leukocyte and platelet counts were also seen for several of our HNSCC patients, and we observed significant correlations between IL6 levels, WBC counts, and CRP levels. However, none of these three parameters differed significantly between the main patient subsets identified in our clustering analyses based on the IL6 family profiles or the IL6 family/IL1 subfamily/TNFα clustering analyses. These observations suggest that the WBC count and the CRP level depend on differences in IL6 levels rather than on the overall cytokine profiles. The increased WBC counts (as well as the increased CRP levels) could be caused by IL6, which is secreted by bone marrow stromal cells (e.g., osteoblasts, mesenchymal stem cells), and IL6 is an important regulator of normal hematopoiesis and a stimulator of myelopoiesis during inflammation [76]. However, increased levels of other hematopoietic growth factors during inflammation may also contribute to the increased WBC counts [77]. Previous studies are also consistent with the hypothesis that different phenotypic characteristics of the acute phase reaction may reflect the effects of different drivers/inducers [1].

When analyzing our overall data, we observed an association between good prognosis and HPV positivity, whereas adverse prognosis was associated with increasing both CRP levels and WBC counts. These are expected associations that have been described in previous studies [9,21,22,23,47,48,49,50,51,52,53,54,55,56,57,78] and therefore suggest that our patient cohort is representative. Furthermore, histopathological features like tumor lymphocyte density predict prognosis both in HPV-negative and HPV-positive oropharyngeal cancer patients [79]. Thus, it would definitely be interesting to study possible associations between tumor infiltration of various immunocompetent cell subsets (i.e., T cell subsets, natural killer (NK) cells, monocyte subsets) and the acute phase cytokine levels/profile. The expression of immune checkpoint molecules in the tumor microenvironment should probably also be included in such studies.

We were not able to differentiate between the HPV-negative and -positive patients based on the peripheral blood cytokine/receptor or CRP levels at diagnosis. This is to some extent surprising, in that the two cancer diseases have fundamentally different causes, i.e., HPV as a viral factor and patients with HPV-negative tumors in principle being due to DNA mutations most often caused by smoking in combination with alcohol consumption [80]. Furthermore, the overall acute phase cytokine profile was not associated with cancer-specific HNSCC survival. We therefore investigated the possible prognostic impact of single mediators. CRP, IL6, and IL1RA showed strong associations with survival both when investigating the whole patient cohort as well as HPV-negative (all) and HPV-positive (IL6) patient subsets. Particularly, an intriguing observation was the unique survival prediction of the individual parameters CRP and IL6 from the same model; they showed prognostic impact independent of each other.

The levels of inflammatory markers can also be influenced by inflammaging, i.e., a smoldering pro-inflammatory phenotype that can be part of the aging process [81]. We therefore adjusted the IL6/CRP survival predictions found in Cox regression analysis by this parameter without finding any fundamentally changed result. HNSCC patient survival was also suggested to be associated with gp130, IL31, and IL6Rα (HPV negative), as well as indicated by CRP and CNTF (HPV positive). Taken together, these observations show that cytokine/cytokine receptor levels likely reflect different aspects of the cancer-associated systemic acute phase cytokine response with one common dimension associated with IL6, whereas another HPV-negative-related dimension is found regarding gp130, IL31, IL6Rα, and IL1RA levels.

It should also be noticed that IL1RA is an acute phase marker and high levels have previously been associated with aggressive disease and/or adverse prognosis in various malignancies, including T cell lymphoma [82], sarcoma [83], colorectal cancer [84,85], and thyroid cancer [86]. Our present study is the first to suggest that this is true also in HNSCC.

In the HPV-negative patient group, IL6, gp130, and IL6Rα showed prediction of survival for the HNSCC disease. IL6Rα, demonstrating impaired prognosis, may suggest that the IL6 prognostic impact is predominantly mediated through trans signaling in these patients. On the other hand, elevated levels of the decoy receptor gp130 possibly reduce this argument. Future studies are needed in order to elucidate this question.

Both our clustering analyses identified patient subsets that showed increased levels of several mediators, but CNTF differed from these mediators and showed decreased levels for these patients. CNTF can be detected in serum/plasma in both healthy and disease states and can be released by various cells (e.g., astrocytes, osteoblasts, osteoclasts, chondrocytes); its receptor is detected in various tissues (e.g., skeletal muscle, kidney, liver, lung, bone marrow, bone/osteoblasts) and can be cleaved from the cell membrane and thereby be responsible for trans signaling [87,88,89]. However, CNTF can bind not only to the CNTF receptor but also to the LIF receptor, and through these receptor bindings it can activate several downstream intracellular pathways [28,90]. CNTF can induce an acute phase response [91], but the low levels in the HNSCC patients suggest that this effect is less important in these patients. However, it is an important metabolic regulator both at the cellular level (e.g., bone, adipocyte) [92] but also at the systemic level, where it seems to alter glucose, lipid, and amino acid metabolism [29]. It is important both for immunoregulation and regulation of hematopoiesis. Further studies are needed to clarify the mechanisms that are important for its potential prognostic impact in HPV-positive HNSCC.

Although the study comprised a substantial number of HNSCC patients, some of these have a short observation time since diagnosis, and measurements targeted at survival may then be of limited sensitivity. This applies in particular to the HPV-positive patients because of their excellent prognosis. Moreover, a conservative approach was used in the statistical analyses. For this reason, negative results should be treated with some caution. This may be regarded as a weakness of the present study, but it is the first study of the mediator profile as a prognostic parameter in human HNSCC, and our priority has therefore been a careful analysis and interpretation of the results. On the other hand, the present positive results should be well substantiated and call for additional studies, especially along clinical dimensions. Only patients to be treated by curative intent were included. Including all newly diagnosed patients would likely have moved especially the HPV-negative group to comprise additional patients more strongly affected by the disease, consequently with a more serious prognosis and possibly also substantial inflammation present. Our results thus apply to the HNSCC disease before the end stage of the cancer, which is also the biologically most interesting part of the disease process.

ROC analyses permitted the study of survival relative to cytokine/receptor and CRP levels concerning clinical cut-off values. According to the present material, low mediator values could define a cut-off value with an 80% sensitivity of disease-specific death, and a higher (but still relatively low) value provided a cut-off value with an 80% specificity for cure. Such easily available information holds promise to be prognostic characteristics and demonstrate the clinical implications, as it may be important for treatment decision-making in HPV-negative HNSCC. Our present study is the first to suggest a prognostic impact of the systemic inflammatory profile in HNSCC patients. Further studies including a larger number of patients and alternative mediator combinations should therefore be encouraged to verify our present results and to identify the optimal combination of mediators for prognostication. Systemic levels of inflammatory mediators have been included in prognostic indices together with other (e.g., clinical) parameters for various inflammatory conditions, including malignant diseases [62]. Our present study and our ROC analyses suggest that relevant cut-off levels can be identified and can then be used to design a prognostic index for patients with HPV-negative HNSCC, but the number of patients in our present study is too low to allow a suggestion of the exact cut-off values to be used in such a prognostic index.

The half-life of a cytokine in circulation is apparently only a few minutes [93], and therefore the blood cytokine concentration may vary substantially. Thus, basing a conclusion on one sample drawn from each patient should be done with caution. On the other hand, the fact that we and others have shown, e.g., prognostic valid information collected from studies with similar design, adds value to the concept [20].

HNSCC generally show a stronger infiltration of immunocompetent cells than most other malignancies, including both various T cell subsets, NK cells, and monocytes [94,95,96]. However, the degree of T cell infiltration seems to vary between patients, but for most patients, the infiltrating T cells seem to include a majority of T cells with an effector memory phenotype; the fraction of naïve T cells seems to be decreased, whereas the fractions of various regulatory T cells seems to be increased [95]. This study also showed that infiltrating T cells seem to have a relatively high expression of immune checkpoint molecules compared with circulating T cells. Both HPV-positive and HPV-negative tumors show infiltration of various T cell subsets, NK cells, and monocytes [96].

The prognostic impact of tumor infiltration in HNSCC has been investigated in several previous studies. Tumor-infiltrating immunocompetent cells seem to have a prognostic impact for both HPV-positive and HPV-negative patients [79]. First, strong infiltration of CD3^+^ T cells has been associated with prolonged survival in several studies; this favorable prognostic impact seems to be mediated especially by CD8^+^ T cells, but a low frequency of regulatory T cell infiltration may also contribute. A recent study suggested that CD4^+^ T cells may also contribute to the more favorable prognosis [94,97,98]. Second, high infiltration of NK cells has also been associated with a favorable prognosis [96]. Third, total monocyte infiltration did not show any significant impact on prognosis in one of the previous studies [94], but the prognostic impact of monocytes may differ between patients and depend on the relative frequencies of various monocyte subsets in the tumor and their molecular interactions with the cancer cells, especially their CD47 expression [99]. Finally, the degree of PD-L1 expression seems to have a very limited (if any) prognostic impact [100].

An important question is whether the altered systemic levels of acute phase cytokine mediators are caused by tumor-infiltrating immunocompetent cells, thereby reflecting this local immunological response event rather than systemic inflammatory responses. As described above, strong tumor infiltration of various immunocompetent cell subsets has a favorable prognostic impact and is associated with prolonged survival. In contrast, as can be seen by the hierarchical clustering analysis presented in Figure 6, improved survival was observed especially for patients with relatively low levels of acute phase cytokine mediators. Thus, strong tumor infiltration and high systemic acute phase mediator levels have an opposite impact on survival, and taken together, these observations suggest that increased levels of acute phase cytokine mediators reflect a systemic inflammatory response and not a local immune-mediated anticancer effect. This hypothesis is further supported by previous studies describing significant associations between an adverse prognosis and high peripheral blood levels of activated T cells and monocytes in patients with HNSCC [101,102].

The fact that the immune system is apparently important to the prognosis of HNSCC patients adds to immune modulation as a potential new dimension of treatment, which has already been approved regarding cytotoxic T-lymphocyte-associated protein (CTLA)-4 and programmed cell death protein (PD)-1 inhibition as an option for head and neck cancer patients [103]. The present results also suggest targeting the inflammatory dimension of HNSCC in treatment with agents, such as, e.g., specific antibodies. In any case, further studies concerning carcinoma immunology interactions are highly needed.

## 5. Conclusions

To conclude, CRP levels reflect only a part of the cancer-associated inflammation in patients with head and neck squamous cell carcinoma, and other acute phase cytokine mediators should be further investigated, both as possible prognostic biomarkers in these patients, as well as to improve the understanding of carcinogenesis in this malignancy.

## Figures and Tables

**Figure 1 biomedicines-08-00418-f001:**
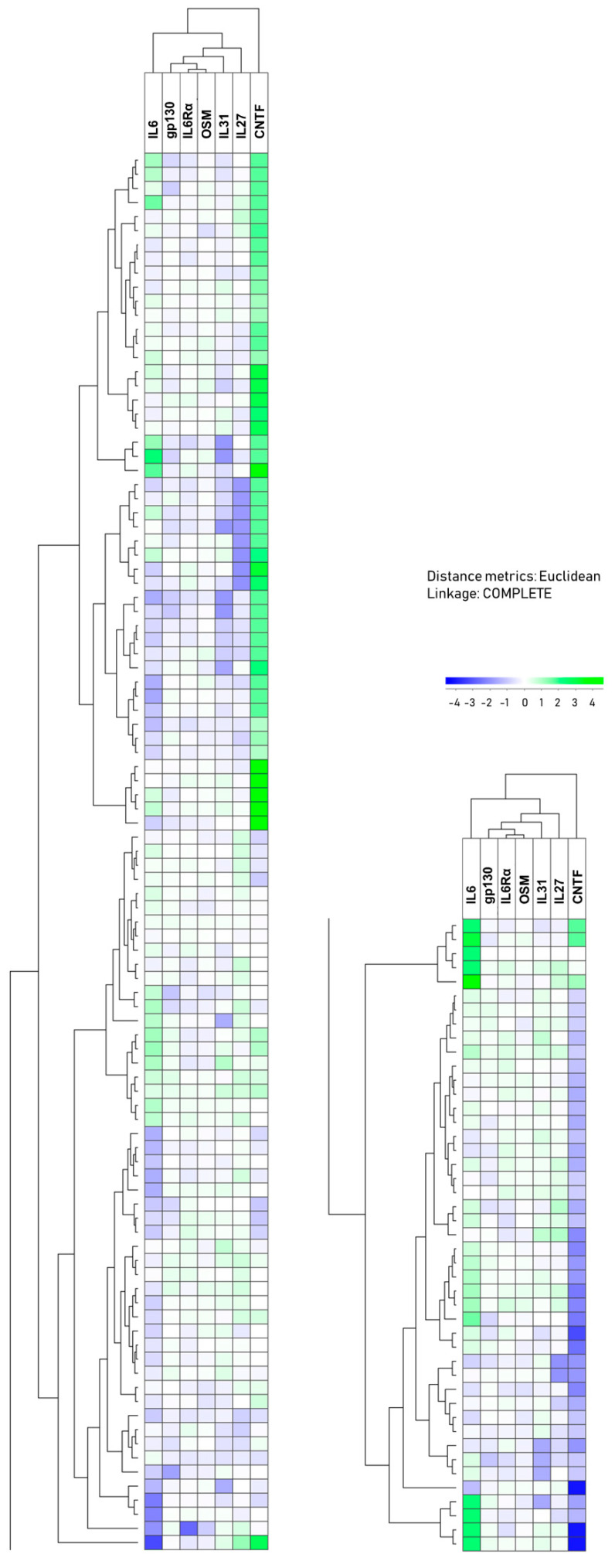
The plasma profile of IL6 family cytokines in head and neck squamous cell carcinoma patients; a hierarchical clustering analysis. The analysis included the seven soluble mediators IL6, gp130, IL6Rα, oncostatin M (OSM), IL31, IL27, and CNTF. The cytokine mediators are indicated at the top of the figure and the patient clustering to the left sides. The analysis created two main clusters (represented by the two columns) including 99 and 45 patients, respectively. The presentation of these data is explained by the middle bar above the right column cluster. Clinical and biological characteristics of the individual patients can be found in Appendix A.

**Figure 2 biomedicines-08-00418-f002:**
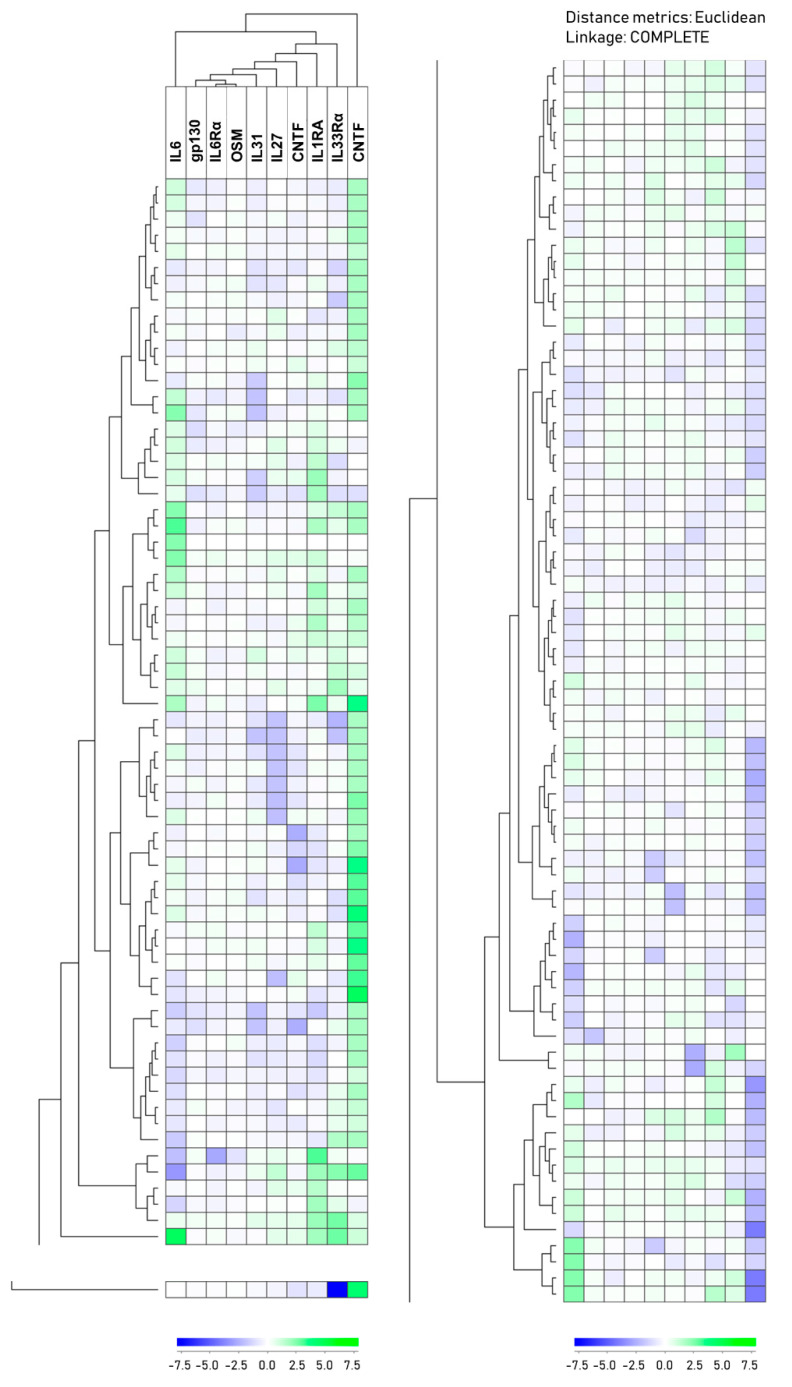
The plasma profile of acute phase cytokines in HNSCC patients; a hierarchical clustering analysis including IL6 family cytokines (IL6, IL6Rα, gp130, IL27, IL31, OSM, CNTF), two IL1 cytokine family mediators (IL1RA, IL33Rα), and TNFα. The cytokine clustering is indicated at the top of the figure and the patient clustering to the left sides. Cluster data are explained in the lower part of the figure. This analysis identified an upper main cluster (left part of the figure, 66 patients), a lower main cluster (right part of the figure, 77 patients), and an outlier located at the bottom of the cluster diagram (see lower left in the figure). Appendix A shows the clinical and biological characteristics for each patient.

**Figure 3 biomedicines-08-00418-f003:**
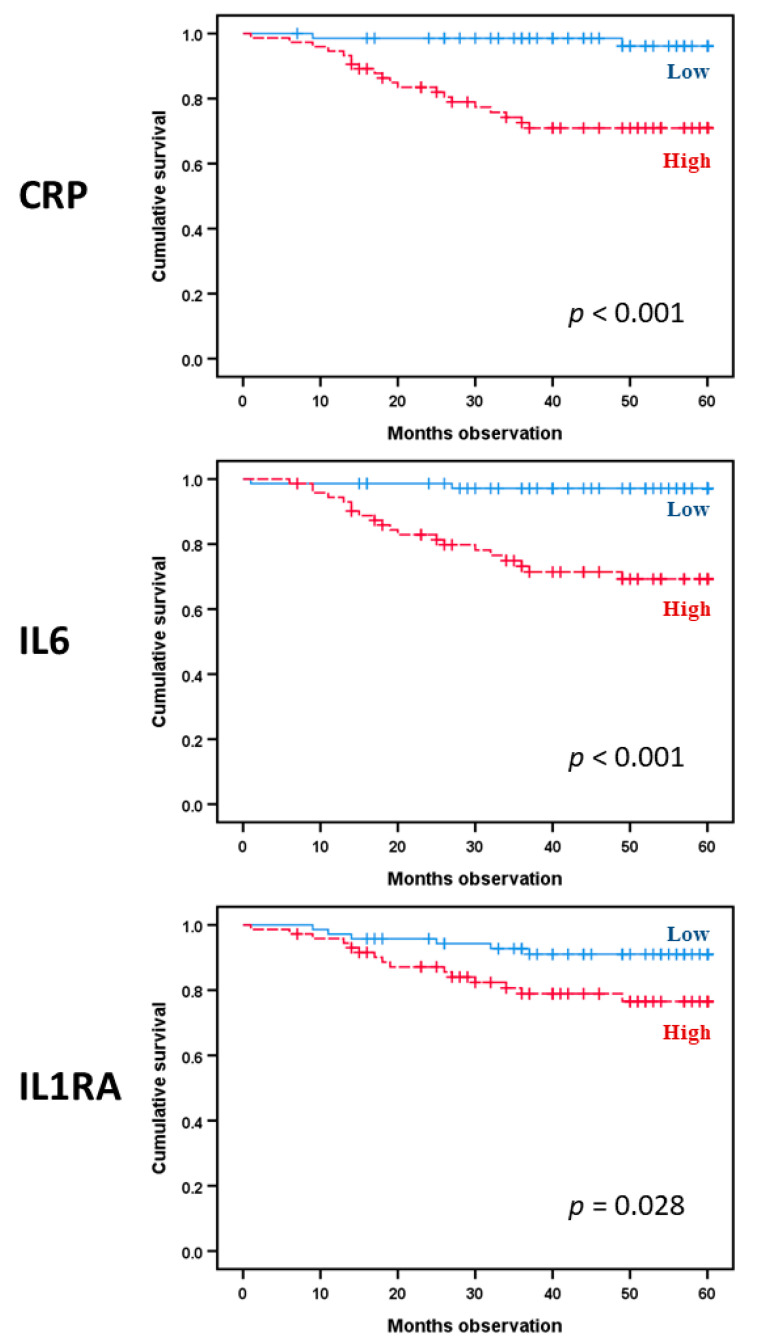
Disease-specific survival prediction from CRP, IL6, and IL1RA; a comparison of high and low scorers as defined by the median mediator level in 144 patients with head and neck squamous cell carcinoma.

**Figure 4 biomedicines-08-00418-f004:**
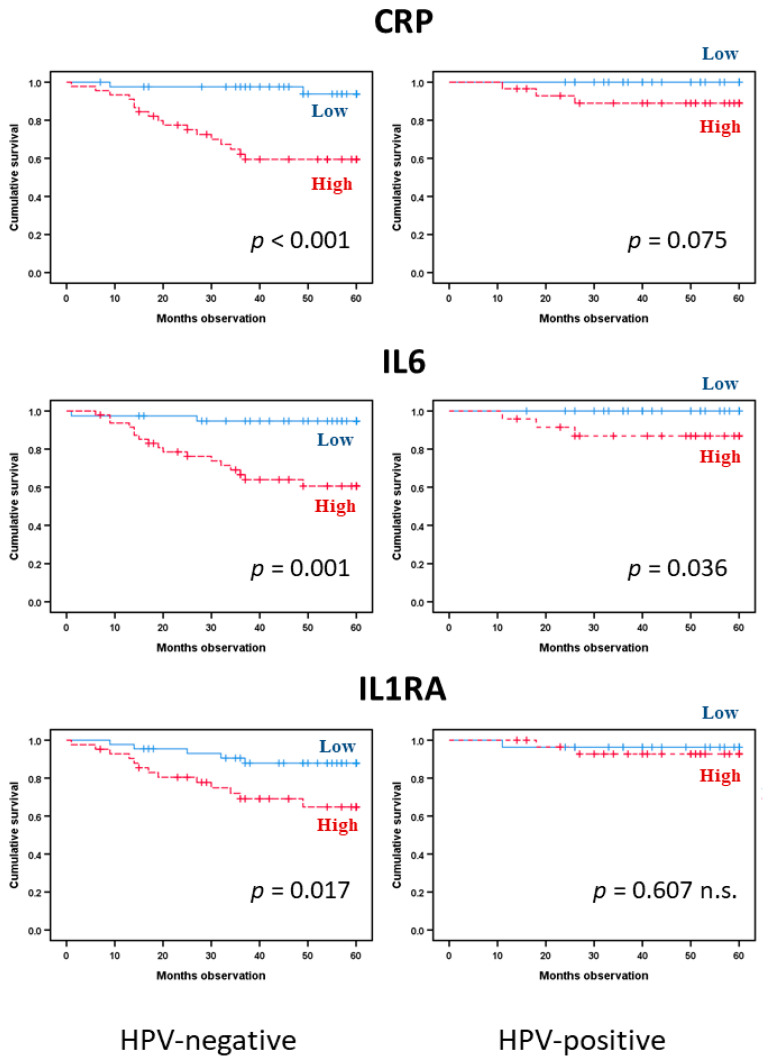
Comparison of HPV-negative and HPV-positive HNSCC patient survival. Prediction from the dichotomized by median value mediators CRP (upper), IL6 (middle), and IL1RA (lower row) comprising 144 patients. The *p*-values are indicated in the figure (log-rank tests).

**Figure 5 biomedicines-08-00418-f005:**
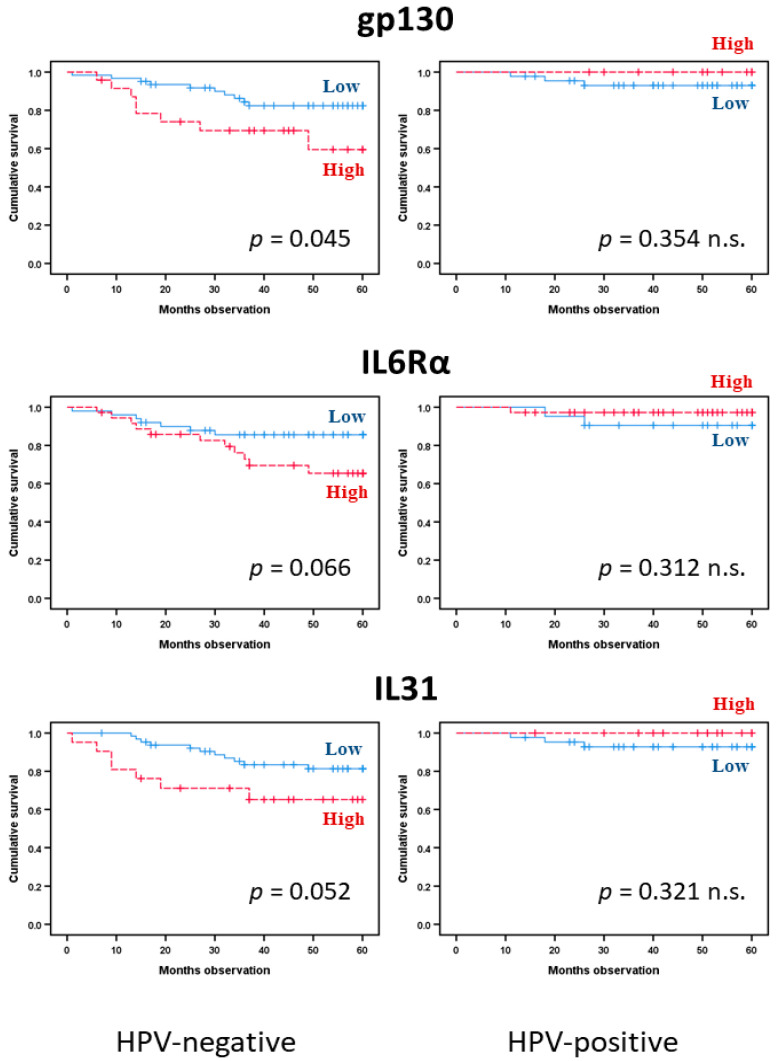
Kaplan–Meier curves demonstrating HPV-dependent prediction of survival in 144 head and neck squamous cell carcinoma patients. glycoprotein130 (gp130, upper), IL6Rα (middle), and IL31 (lower panel) levels were divided by quartiles and classified into high or low. The *p*-values are indicated in the figure (log-rank tests).

**Figure 6 biomedicines-08-00418-f006:**
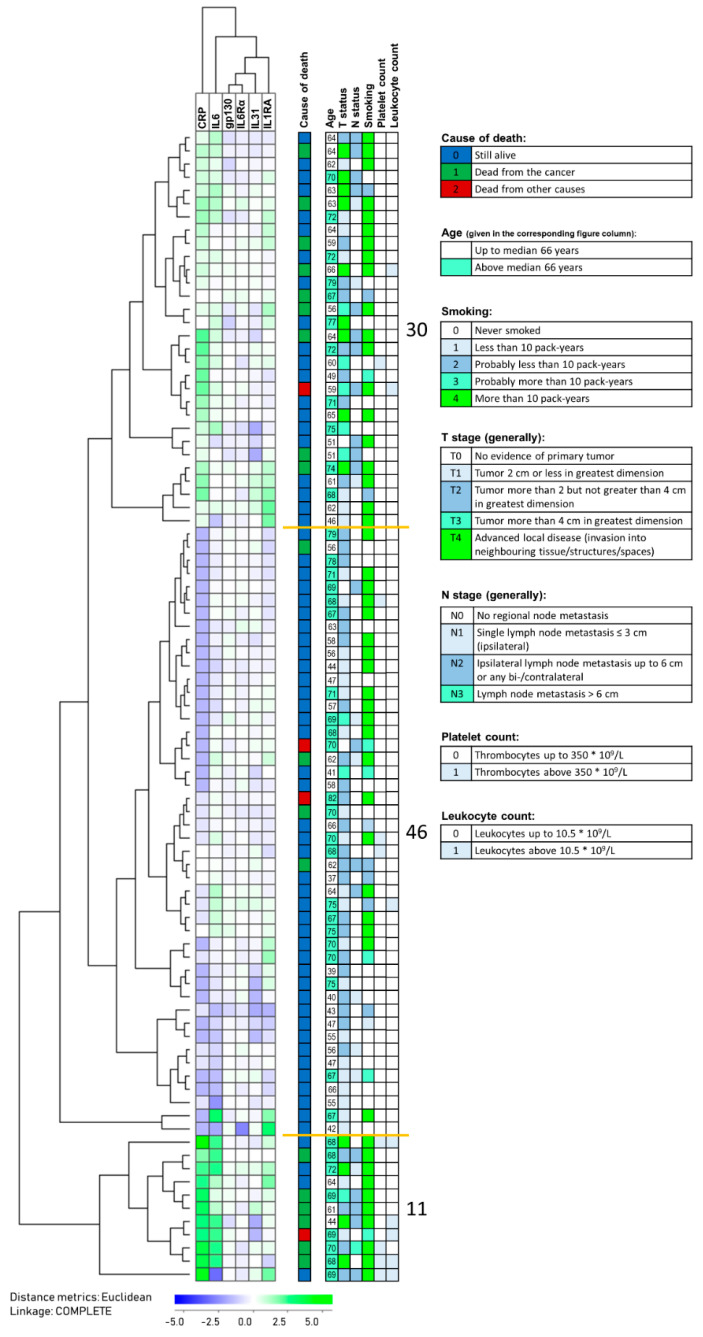
An unsupervised clustering analysis for HPV-negative patients based on six acute phase biomarkers that showed significant or borderline associations with survival in Kaplan–Meier analyses (Figure 4 and Figure 5). The cytokine clustering is indicated at the top of the figure and the patient clustering to the left. Cluster data are explained in the lower part of the figure. The isolated middle column indicates patient state by the end of the study; still alive (blue), dead from HNSCC (green) or other cause (red), and further complementary information including available histopathological findings, as well as inflammatory markers [62]. The number of patients in each of the three main clusters are indicated to the right.

**Figure 7 biomedicines-08-00418-f007:**
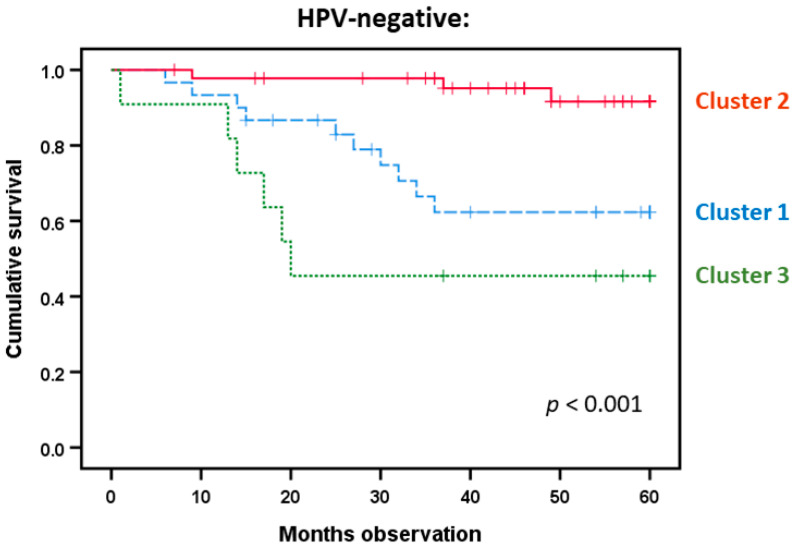
Survival analysis of HPV-negative patients; a comparison of the patient clusters identified in Figure 6. This subset identification was based on the systemic levels of the six acute phase mediators IL6, gp130, IL6Rα, IL31), and the IL1 subfamily member IL1RA. The different lines represent the three identified clusters (n = 30, n = 46, and n = 11, respectively). The *p*-value is shown in the figure.

**Figure 8 biomedicines-08-00418-f008:**
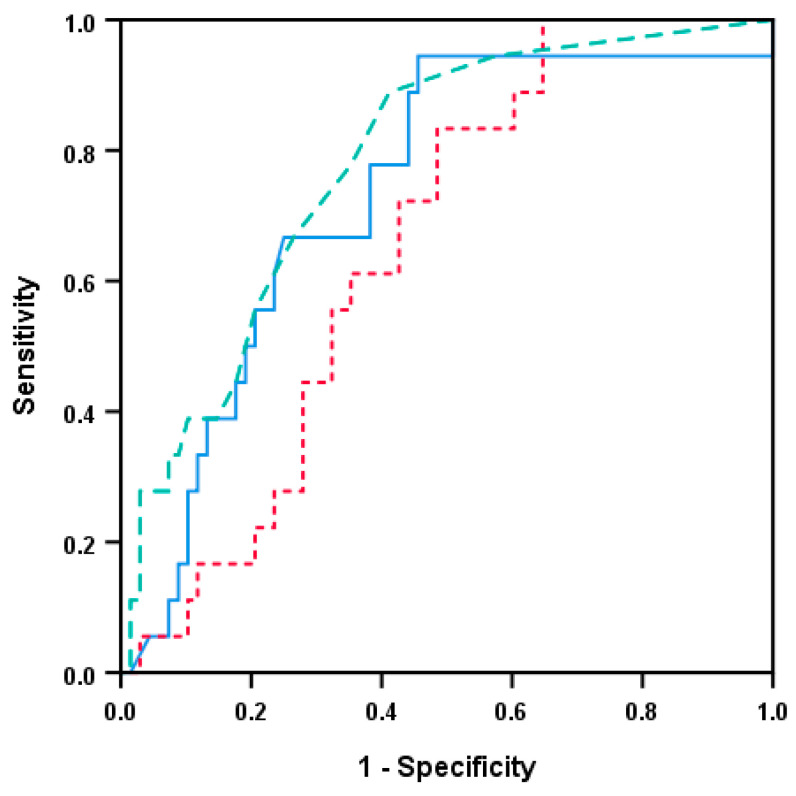
Receiver operating characteristic (ROC) curve based on previous significant systemic values from HPV-negative HNSCC patients; a comparison of CRP, IL6 (both n = 87), and IL1RA (n = 86). Blue continuous line denotes IL6, green semi-hatched line represents CRP, whereas IL1RA is presented by the red dotted line. The respective areas under the curve (AUC) and corresponding significance were CRP: AUC 0.733, *p*-value 0.002; IL6: 0.780, *p* < 0.001; and IL1RA 0.653, *p* = 0.047.

**Table 1 biomedicines-08-00418-t001:** Clinical and biological characteristics of the HPV-negative versus HPV-positive head and neck cancer patients included in the study (n = 144).

PARAMETER	HPV-Negative or Unknown (n = 87) ^1^	HPV-Positive (n = 57)	*p*-Value (Chi-Square Test)
Age at diagnosis (median and range, years)	66.0 (37–82)	56.0 (38–78)	0.062
Sex (males/females, numbers)	59/28	46/11	0.089
Blood values (median and range) ^2^			
Hemoglobin (g/100 mL)	14.5 (9.7–17.7)	14.9 (11.5–17.0)	0.782
Leukocytes (×10^9^/L)	7.4 (4.0–21.1)	6.7 (4.1–15.2)	0.497
Thrombocytes (×10^9^/L)	255 (112–405)	254 (157–759)	0.421
CRP (mg/L)	3 (1–150)	3 (1–54)	0.698
Erythrocyte sedimentation rate (mm/h)	14 (2–106)	13 (1–83)	0.309
Serum-glucose (mM)	5.6 (3.5–14.1)	5.9 (4.0–15.2)	0.261
Serum-creatinine (μM)	76 (47–167)	79 (48–135)	0.273
Tumor localization			< 0.001
Tonsils, tongue base, head and neck cancer origo inserta	5 (5.7)	50 (87.8)	
Other oropharyngeal sites	12 (13.8)	4 (7.0)	
Oral cavity	61 (70.1)	1 (1.8)	
Other (nasopharynx, hypopharynx, glottis)	9 (10.3)	2 (3.5)	
Cancer characteristic			
T stage (number and percentage)			0.431
T0/unknown primary	3 (3.4)	4 (7.0)	
T1	29 (33.3)	13 (22.8)	
T2	34 (39.1)	29 (50.9)	
T3	8 (9.2)	5 (8.8)	
T4	13 (14.9)	6 (10.5)	
N stage (number and percentage) ^3^			< 0.001
N0	57 (65.5)	11 (19.3)	
N1	9 (10.3)	13 (22.8)	
N2	20 (23.0)	31 (54.4)	
N3	1 (1.1)	2 (3.5)	
M stage	All patients M0	All patients M0	-
Primary treatment			
Surgery including neck dissection, numbers Primary, primary surgery, Neck, neck dissection) ^4^	Primary 64 Neck 34	Primary 10 Neck 8	Primary < 0.001 Neck 0.001
Radio-/chemotherapy ^5^, numbers	62/23	56/46	<0.001

^1^ The HPV status was not determined for 42 patients; all except one of these patients had primary tumor localized outside the oropharyngeal region and these tumors were therefore grouped together with the HPV-negative tumors [58]. ^2^ Erythrocyte sedimentation rate was only available for 139 (84/55) patients and glucose levels for 141 (85/56) patients. ^3^ Lymph node metastasis according to the Union for International Cancer Control, seventh edition (2009). ^4^ Surgical treatment was either resection of the primary tumor, resection of the primary tumor combined with neck dissection, or only neck dissection if the site of the primary tumor was not known (including consideration for further intervention). ^5^ The current treatment for HPV^+^ pharyngeal tumors is radio(chemo)therapy alone. Surgery may follow if there is evidence of residual.

**Table 2 biomedicines-08-00418-t002:** Hierarchical clustering analysis based on the plasma levels of seven interleukin (IL)6 family members, a comparison of the two main clusters. Plasma levels of IL6 family members, IL1 subfamily members, tumor necrosis factor (TNF)α, C-reactive protein (CRP), and the normal peripheral blood cell counts in patients with head and neck squamous cell carcinoma; a comparison of the two main clusters identified by unsupervised hierarchical clustering analysis based on seven IL6 family members (Figure 1). The Mann–Whitney U-test was used for statistical comparison between the groups. Due to technical reasons, cytokine mediators were only available for 143 (99/44) patients except for ciliary neurotrophic factor (CNTF) levels, which were available for 142 (98/44) patients. IL6 values were present for all 144 patients.

Mediator	Upper/Left Main Cluster (n = 99)	Lower/Right Main Cluster (n = 45)	*p*-Value
gp130 (pg/mL)	81,943 (25,184–108,957)	83,253 (41,057–101,998)	0.399
IL6Rα (pg/mL)	32,289 (5119–44,259)	↑ 34,839 (21,527–47,290)	0.039
IL6 (pg/mL)	1.66 (0.20–9.41)	↑ 2.50 (0.83–45.03)	<0.001
IL27 (pg/mL)	457 (116–1096)	463 (116–882)	0.402
IL31 (pg/mL)	62.5 (19.4–132.2)	↑ 78.5 (23.8–123.0)	0.003
OSM (pg/mL)	4642 (2487–5799)	4531 (3165–5747)	0.887
CNTF (pg/mL)	1113 (293–13,595)	↑ 206 (33.8–1980)	<0.001
IL33Rα (pg/mL)	16,950 (75.9–107,876)	18,588 (8434–113,011)	0.109
IL1RA (pg/mL)	356 (119–3185)	↑ 468 (194–1409)	0.027
TNFα (pg/mL)	15.8 (2.6–28.6)	↑ 17.5 (2.6–28.6)	0.016
CRP (mg/L)	2 (1–55)	↑ 4 (1–150)	0.003
Hemoglobin (g/100 mL)	14.6 (10.9–17.7)	14.8 (9.7–17.0)	0.545
Leukocytes (×10^9^/L)	6.7 (4.0–15.2)	7.4 (4.0–21.1)	0.276
Thrombocytes (×10^9^/L)	252 (112–441)	261 (128–759)	0.458

**Table 3 biomedicines-08-00418-t003:** Hierarchical clustering analysis based on plasma levels of seven IL6 family members, two IL1 subfamily members (IL1RA, IL33Rα), and TNFα; a comparison of the two main clusters. Soluble levels of IL6 family members, IL1 subfamily members, TNFα, CRP, and the normal peripheral blood cell counts (lower part) in patients with head and neck squamous cell carcinoma; a comparison of patients in the two main clusters identified by unsupervised hierarchical clustering analysis based on all 10 soluble mediators (Figure 2, one additional patient allocated to upper main cluster due to high CNTF). The Mann–Whitney U-test was used for the statistical comparisons. Due to technical reasons, cytokine mediators were only available for 143 (66/77) patients except for CNTF levels, which were available for 142 (65/77) patients. IL6 values were present for all 144 patients.

Mediator, Median (Range)	Upper Main Cluster (n = 66)	Lower Main Cluster (n = 77)	*p*-Value
gp130 (pg/mL)	76,526 (40,175–108,957)	↑ 83,538 (25,184–107,124)	0.023
IL6Rα (pg/mL)	31,413 (5119–44,523)	↑ 34,288 (21,943–47,290)	0.004
IL6 (pg/mL)	2.10 (0.20–45.03)	1.80 (0.36–10.00)	0.217
IL27 (pg/mL)	415 (116–1096)	↑ 485 (116–882)	0.007
IL31 (pg/mL)	62.5 (19.4–132.2)	↑ 75.7 (23.8–123.0)	0.001
OSM (pg/mL)	4647 (2487–5799)	4603 (3165–5747)	0.786
CNTF (pg/mL)	1980 (293–13,595)	↓ 312 (33.8–881.5)	<0.001
IL33Rα (pg/mL)	17,359 (75.9–113,011)	18,202 (7462–64,650)	0.565
IL1RA (pg/mL)	400 (119–3185)	361 (194–1152)	0.111
TNFα (pg/mL)	14.4 (2.6–26.2)	↑ 18.2 (2.6–28.6)	<0.001
CRP (mg/L)	3 (1–150)	2 (1–39)	0.128
Hemoglobin (g/100 mL)	14.6 (11.4–17.0)	14.8 (9.7–17.7)	0.502
Leukocyte counts (×10^9^/L)	7.0 (4.2–21.1)	6.9 (4.0–16.1)	0.140
Thrombocyte counts (×10^9^/L)	257 (112–759)	250 (128–396)	0.870

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
