# Peer review of "The Acute Phase Reaction and Its Prognostic Impact in Patients with Head and Neck Squamous Cell Carcinoma: Single Biomarkers Including C-Reactive Protein Versus Biomarker Profiles"

_biomedicines, 2020, doi:10.3390/biomedicines8100418_

Round 1

Reviewer 1 Report

the manuscript has been significantly improved

Author Response

We appreciate the reviewer’s comment. New manuscript parts are marked in blue. The discussion have been extended with two subparagraphs, and more information about Regional Ethics approval is also included in the Materials and Methods section.

Reviewer 2 Report

This is an interesting study presenting the prognostic impact of the acute phase reaction-related biomarkers in patients with head and neck squamous cell carcinoma. The manuscript is well-written. A few criticisms need to be clarified as below: 

  1. After hierarchical clustering analysis, each cytokine mediator's confidence interval is relatively wider (page 10). Besides, most of the statistical analyses such as Kendall's tau and Mann-Whitney U-tests are nonparametric tests, which might raise the uncertainty. I would suggest the authors address the potential weaknesses of this study as their study limitation.
  2. By using the receiver operating characteristic (ROC) curve, will it be possible to analyze further and provide the cut-point value for each mediator involved in patient’s survival with HPV-negative?

Author Response

We are grateful for the two comments made by the reviewer. We have now extended the discussion and addressed these two comments in two of the chapters in the Discussion section. The new parts are marked with blue. Information about Regional Ethics approval is included in the Materials and Methods section.

This manuscript is a resubmission of an earlier submission. The following is a list of the peer review reports and author responses from that submission.

Round 1

Reviewer 1 Report

The authors  investigated the systemic levels of IL6 family mediators (gp130, IL6Rα, IL6, IL27, IL31, OSM, CNTF), IL1 subfamily members (IL1RA, IL33Rα), and TNFα in HNSCC patients.

Specific points:

  • It is not clear the total number of males and females. Please specify in the section 2.1. I find some inconsistencies between the main text ( line 94-98) and the number of HPV negative and positive presented in the table 1. Please explain better or correct.
  • Section 2.2. please specify how the authors calculated the cytokine levels for the cancer patients. Real time PCR? Please add this information and add more detail about the methodology.
  • Section 2.2. How many cancer samples? How many healthy individuals? The authors should indicate the precise numbers of samples for each group.
  • Section 2.3. line 121.what is CPR? Please define
  • Section 2.4. hierarchical clustering shows that the patients can be divided in two main subsets. please specify the total number of patiens. I think 144? And the number of samples for each subset. Furthermore, it is not clear how the authors defined that IL6, IL31, CNTF, and CRP are able to cluster in a higher significant difference the two subsets than IL6Rα, IL1RA, and TNFα. Please explain better. I don’t see for example in the figure 1 TNFa and CRP. Why?
  • The two main subsets obtained in the figure 2 are the same of the figure 1? What is the difference? Please explain better the differences of analysis between section 2.4 and 2.5.
  • It is not clear the correlation between HPV and HNSCC. Please discuss other works that found this correlation.
  • Please add more details about the methodologies used in the study

Reviewer 2 Report

In this paper, authors showed the proteins of acute phase reaction have prognostic value of Head and neck squamous cell carcinoma (HNSCC) patient. They showed that gp130, IL6Ra and IL31 were correlated with prognosis in HPV negative HNSCC. Furthermore, authors showed that IL6 was correlated with prognosis in HPV positive HNSCC. Although I feel that this paper contains interesting findings, the findings as reported are insufficient novelty and not suitable to permit publication in Cancers.

I have several comments as the following:

  1. Authors should analyze the association between acute phase reaction and histopathological findings like differentiation, lymph nodes metastasis and inflammation in primary tumor.
  2. Authors should examine the pathway analysis by using acute reaction factors which had prognostic value. It is interesting whether these factors are involved in same pathway or not.

Reviewer 3 Report

This is an interesting study presenting the prognostic impact of the acute phase reaction-related biomarkers in patients with head and neck squamous cell carcinoma. The manuscript is well-written. Only a few criticisms need to be clarified as below: 

  1. After hierarchical clustering analysis, each mediator's confidence interval is relatively wider (page 9), and most of the statistical analyses such as Kendall's tau and Mann-Whitney U-tests are nonparametric tests, which might raise the uncertainty. Is it a considered study limitation?
  2. By using the receiver operating characteristic (ROC) curve, will it be possible to analyze further and provide the cut-point value for each mediator involved in HPV-negative patients' survival?